# Examining the Impact of Corporate Governance on Investors and Investee Companies: Evidence from Yemen

**Fahd Alduais \***, **Jafer Alsawalhah** and **Nashat A. Almasria \***

Department of Accounting, Philadelphia University, Jarash Road, 20 KM, Amman 19392, Jordan
* Correspondence: falduais@philadelphia.edu.jo or accofahd@hotmail.com (F.A.); nalmasria@philadelphia.edu.jo (N.A.A.)

**Abstract:** The purpose of this study was to determine whether corporate governance is an important and effective technique for enhancing investors' confidence in existing and prospective companies and for creating opportunities for safe investment in Yemen. A survey was conducted among certified public accountants to assess the importance of corporate governance. We employed regression analysis to test our hypothesis. According to the results of the study, corporate governance is an essential component of success for companies, and those firms that apply corporate governance best practices are highly regarded. Additionally, the findings suggest that regulators, policymakers, and standard-setters should raise awareness of the importance of protecting shareholders' rights by providing seminars and courses for Yemeni media, unions, and professional associations. Moreover, in an environment of uncertainty there is a reluctance to invest and a prevalent tendency to invest in real estate. Furthermore, the results indicate that corporate governance is not practiced by all companies but only to a limited extent by some joint-stock companies. Most of the Yemeni companies that have adopted CG are joint-stock companies, so investors prefer to invest in these companies. The findings of this study provide valuable insights for regulators, practitioners, and academicians. We recommend that this survey be extended to a larger sample, including supervisory managers of companies. This study provides an insightful contribution, because it clarifies the importance of corporate governance for Yemeni investors and investee companies.

**Keywords:** transparency; awareness; disclosure; investee companies; investor protection; corporate governance in Yemen

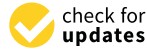



## 1. Introduction

The concept of corporate governance (CG) has emerged in recent years as a mechanism for economic development, including procedures and the management of related parties in organizations. It has received significant attention, especially from supervisory and control authorities. CG emerged because of the shortcomings that permeated the laws and legislation that govern business and commercial activities, which led to many cases of bankruptcy, insolvency, and financial hardship for many large companies, affecting thousands of shareholders and capital owners. Managers often do not align their interests with shareholders due to the fact of information asymmetry.

The function of CG is to mitigate agency problems that arise between related parties to create shareholder value. These concerns and interests are related to how the investors monitor the company and attempt to avoid the risks and costs of investing. Studies have been conducted to investigate these issues (Kling and Gao 2008; Al-hilu et al. 2017; Healy and Palepu 2001; Habib and Jiang 2015; Hirshleifer 2001; Shleifer and Vishny 1997; Tam 2000; OECD 2004; Coombes and Watson 2000; Elbadry et al. 2015; Fosberg 2004). Some studies have been conducted on Yemeni banks, which play a role in CG, and the majority are joint-stock companies (Alobaidi et al. 2017; Basuony et al. 2014; Alawi et al. 2016; Al-Baidhani 2018; Al-Matari et al. 2016; Al-Homaidi et al. 2020a; Ismal et al. 2014; Ali Saleh

Al-magharem et al. 2019; Qaid and Alhamidi 2020). Previous studies have also found that firms with good CG perform better than firms with less CG (Larcker et al. 2007), and firms with better CG have shown better performance (Attig et al. 2013). From a theoretical perspective, firms with good CG are expected to have a better competitive advantage than firms with poor CG.

In order to anticipate opportunistic behavior, CG is supposed to reduce information asymmetry and ally management with shareholders' interests (Menshawy et al. 2021; Guluma 2021; Holm and Schøler 2010). According to agency theory, management entrenchment, free cash flow, overconfidence, and the higher echelon method are a few models that can be used to explain wasteful investment (Menshawy et al. 2021). CG is a crucial instrument for stopping opportunistic managerial behavior, according to agency theory (Hlel et al. 2020). Corporate governance is a set of control systems that looks out for the interests of investors (Utama et al. 2017; Alduais et al. 2022a). Efficient investment, according to Biddle et al. (2009), refers to management selecting investments with high returns. On the contrary, excessive or insufficient investment results in inefficient returns. With the former, management selects an investment model with a negative return, whereas in the latter, management terminates investment projects with a high return (Bimo et al. 2022; Ullah et al. 2020).

Conceptually, corporate governance seeks to ensure that the interests of management and investors are aligned (Jacoby et al. 2019). This convergence of interests can lessen information asymmetry which, in turn, lowers the possibility of moral hazard and adverse selection (Suman and Singh 2020; Jacoby et al. 2019). Based on agency theory, moral hazard and adverse selections due to the fact of agency conflicts that arise because of information asymmetry may lead to unproductive investments (Ullah et al. 2020).

Reducing knowledge asymmetry can improve financial data transparency (Zimon et al. 2022). Because management selects assets that yield the best returns for the company, investment efficiency is therefore likely to improve (Menshawy et al. 2021). In theory, corporate governance is designed to lessen management's propensity for opportunistic behavior, safeguard the interests of shareholders, align the objectives of agents and principals, and reduce information asymmetry. These goals will ultimately boost investment efficiency (Suman and Singh 2020).

Investor protection is a crucial issue, and it has priorities that must be met. This study sheds light on the infrastructure of CG and analyzes its adequacy and applicability in companies as a guarantee for investors. This problem remains under investigation in the current literature, while calls for further empirical research in this area persist. This study fills the research gap by surveying practitioners who have a good working relationship with most companies in Yemen. The legal form of individual companies, partners, and families is dominant in Yemen. Few companies in Yemen offer shares for public subscription, and they are not mentioned here. Most companies are restricted to the founders, as there is no legal obligation for companies to offer public subscriptions. All of this involves a series of interrelated reasons. The absence of giant companies has led to a lack of interest in governance, and this is the reason for the lack of interest in establishing a capital market.

The aim of this study was to review and analyze the literature to identify the most important factors that inhibit investors' willingness to invest as well as to attempt to answer the following questions: Research question 1: (a) What are investors focusing on in the market, (b) what role do good CG practices play, and (c) what are the most crucial governance factors with greater influence? Research question 2: (a) Do well-governed companies influence investor decisions and (b) will investors be sophisticated enough to form future joint-stock investment companies? Research question 3: (a) Is increasing awareness of CG in Yemen an important matter, (b) how important is it to establish a stock market in Yemen, and (c) what are the related factors that prevent this investment? Research question 4: (a) What causes investors to refrain from investing and establishing companies but invest in real estate? A cross-sectional study was conducted in Yemen in order to address these research questions.

Our paper contributes to the literature by providing direct evidence of the importance of CG to investors and investee companies. We recommend reforming and improving the factors that will help in strengthening the role of CG and applying it more comprehensively, and raising awareness through the relevant professional associations and unions, the media, and educational institutions. We also examine the possibility of establishing a stock market in Yemen to shed light on the importance of stock markets in motivating investors to contribute to joint-stock companies, which could support the national economy. In addition, we clarify the degree of influence of an environment of uncertainty on investment and investors and how to know their future investment direction in light of the uncertain environment. Fourth, this paper fills a research gap in terms of the importance of good CG for investing companies and investors.

The remainder of the paper is organized as follows: Section 2 reviews the relevant literature and hypothesis development. Section 3 presents the materials, sample, and data collection method used in the study. Section 4 presents the results and Section 5 the discussion of our findings regarding CG and investors. In Section 6, the policy implications and recommendations are elaborated, and Section 7 concludes the paper.

## 2. Literature Review and Hypothesis Development

The Organization for Economic Co-operation and Development (OECD) defines CG as a relationship between a company's management and board, government shareholders, and other stakeholders, which involves determining the means to achieve these goals and monitor performance. The experiences of many countries and companies globally and the financial crises that have affected them have made it clear that good CG is an essential element for the healthy and robust growth of regional, national, and international economies.

Corporate governance includes the procedures and methods for managing related parties in companies (OECD 2004; Aguilera et al. 2019; Farah et al. 2021; Almasria 2021). It is a prominent topic that has attracted the attention of capital markets, business organizations, researchers, and international organizations. Corporate governance emerged as a response to the agency problem and a conflict of interest between a company's management, shareholders, and stakeholders (Almasria 2018; Sener and Selcuk 2019; Core et al. 1999; Chung et al. 2022; Almasria 2022b; Suman and Singh 2020; Chen et al. 2017; Habib and Jiang 2015; Alduais et al. 2022a). Furthermore, instability and turmoil have affected some financial markets, as well as international companies, during periods of manipulation of financial statements, lack of corporate transparency, violation of shareholder rights, and the lack of a sound administrative structure capable of allowing shareholders to achieve their goals. Corporate governance is a good guide for companies, especially in balancing conflicts of interest between investors, company management, and other stakeholders. It works to increase investor confidence and the market value of company shares and helps companies obtain international and local financing, especially after the financial shocks that hit global markets in the mid-1990s (Bimo et al. 2022; Feng et al. 2020; Nguyen et al. 2015; Shahid and Abbas 2019; Otman 2019).

### 2.1. Corporate Governance in Yemen

The CG Guide in Yemen, issued in 2010, was considered to represent a partnership between the government and the private sector in an integrated way. It was aimed at improving the business and investment climate to create appropriate conditions for investment and encourage companies to commit to the principles of governance, and was considered a lifeline for them. The guide includes two annexes, the first for banks, due to the economic importance of this sector in general, and the second for family businesses, due to the dominance of this type of company in the Yemeni market. This guide is complementary to Yemen's enforced laws, foremost of which is Commercial Companies Law No. 22 of 1997 and its amendments, executive regulations, and decisions.

In the Middle East's transition economies, interest in CG is not a new phenomenon. It is especially important in these economies, since they lack the long-established (financial)

institutional infrastructure required to address CG issues (Braendle 2013). Intense CG is crucial for the Middle East and North African (MENA) countries as they attempt to boost economic growth, strengthen competitiveness, and build prosperous societies (Otman 2019). CG is still a slow process in Yemen compared to most MENA countries, including Egypt and the UAE. Implementing a CG system requires a long process divided into stages, starting with raising people's awareness of the need for such a system and explaining its characteristics, and including the need to monitor the system and ensure that it works as planned and the need to ensure that journalists write about the positive and negative aspects of private companies because of their role in economic change in Yemen.

A number of empirical studies have increasingly confirmed that companies character-ized by good governance enjoy higher market valuation (Gompers et al. 2003) and higher market response (Gao et al. 2019) and have a positive effect on the quality of environmental disclosure and performance (Giannarakis et al. 2019; Hong et al. 2021; Almasria 2022b). These studies have generally found that companies with better governance achieve better overall revenue, especially in volatile markets. Investors are increasingly considering com-panies' CG practices when making investment decisions. Moreover, fixed-income investors may accept lower interest rates and longer credit maturity periods. As a result, this may mean a willingness to invest more in companies they consider to be highly governed.

### 2.2. Corporate Governance and Investment

The World Bank's Doing Business 2020 report shows a low minority investor protection score for Yemen of 26.0, ranking 162nd. Comparing the level of investor protection with other good practices and selected economies, Yemen was one of the lowest (World Bank 2020).

Yemen occupies a lower rank in the investor protection index because its companies fail to adhere to CG principles and rules. CG is an essential and significant part of economic change, but The CG is not given adequate attention in Yemen with regard to the economic aspect of the country (see Table 1), based on the survey respondents' answers regarding how businesses adopt governance. In addition, the lack of CG in the business community is considered to be an economic imbalance and is directly reflected in the productive structure of the national economy. It has led to a weak ability to attract private investments, leading to the low competitiveness of the private sector. Few studies have examined investors' attitudes regarding the importance of CG arrangements when making investment decisions by reading about the state of companies in the market; for example, some studies have found that better CG is associated with a higher corporate market valuation and that governance reforms have resulted in a higher corporate market valuation for those companies (Balasubramanian et al. 2010; Klapper and Love 2004; Durnev and Kim 2005) and a strong correlation between governance and market value (Black et al. 2006).

**Table 1.** Doing Business 2020: protecting minority investors' score.

| Region | Rank | Score (0–100) |
|---|---|---|
| Oman | 88 | 56.0 |
| Jordan | 105 | 50.0 |
| Iraq | 111 | 46.0 |
| Iran, Islamic Republic | 128 | 40.0 |
| Yemen, Republic | 162 | 26.0 |

Source: World Bank (2020).

Shareholders and investors trust that the money they invest will not be misused by managers, board members, or majority shareholders. These funds being used in an optimal manner that considers their interests is among the most critical factors in the emergence and development of capital markets (OECD 2004). Boards of directors, managers, and the majority of shareholders can make decisions that achieve their interests at the expense of the interests of other shareholders. However, managers often tend to act in their best interests (Shahid and Abbas 2019; Jensen and Meckling 1976). Other studies report that

managers do not always make investment decisions in the interest of shareholders (Baker and Gompers 2003; Dong et al. 2007; Su 2004).

In light of this, it is vital to appreciate how investors use CG in their investment decisions, which features of CG are the most important to them, and if these features differ according to CG factors, company size, ownership structure, or industry. On a conceptual level, if CG is crucial to any group of investors, then that group of investors should be able to overcome teamwork and information asymmetry concerns. Studies have proven that CG is important to these investors (Ghabri 2022). They indicated that increasing investor protection strengthens CG and improves company performance. Nevertheless, no studies have focused on investors and companies in Yemen and revealed the CG factors that investors look at when making investment decisions (Mihail and Dumitrescu 2021; Gennaro and Nietlispach 2021; Choi et al. 2020; Rehman 2021).

One of the requirements of CG is to increase the level of transparency in order to achieve market efficiency (Farah et al. 2021). Since CG has a policy of guaranteeing shareholder rights, such as the right to participate in company decisions and cast a vote at general company meetings, information should be provided regarding the company's rules and procedures, including voting procedures, and the requirement to disclose financial information. According to Alduais (2022), corporate governance affects the financial market by affecting companies in terms of trading volume and stock prices and enhancing confidence among various parties.

By implementing the rules of CG, stakeholders and shareholders are protected from unfair practices, and administrations are accountable in a manner that protects them (Gerged and Agwili 2020; Guluma 2021). Thus, there will be an increase in investment and a stimulated flow of investment, as well as an increase in national savings, which will maximize profitability and create new employment opportunities. Furthermore, CG strengthens the role of the capital market, mobilizes savings, raises investment rates, and protects minority small investors' rights. This allows the private sector to thrive, remain competitive, finance projects, and become profitable (Spanos 2005).

The process of disclosing financial information is important in reducing the cost of capital and ensuring its continuity in business performance (Alduais 2016, 2019; Alduais et al. 2022b), as CG contributes to attracting foreign and local investments and helps reduce capital flight and combat financial and administrative corruption (Alasbahi 2021; Alduais et al. 2022a). Good corporate governance increases disclosure and transparency and provides financial information that helps improve performance and diversify corporate investments, which increases rates of return on investments (Larcker et al. 2007). There is no doubt that corporate governance is an approach that leads to reviving the economy and raising the efficiency of markets by protecting national investments and giving investors confidence in the state system, which makes it more attractive to foreign direct and indirect investments.

### 2.3. Corporate Governance Factors That Influence Investment Decisions

Along with an increased focus on investment opportunities, there has been an increased interest in CG considerations. A comprehensive review of this research revealed a significant information gap, with little in-depth research on the role of CG in investment decisions. The survey addressed this information gap, focusing on specific CG issues of interest to investors in Yemen. In particular, companies in Yemen may face greater expropriation concerns due to their ownership structure. Thus, one might expect investors not to rely on the same types of CG features as in developed markets (Otman 2019; Farah et al. 2021) but perhaps only some of them or completely other ones. In addition, for funds invested in Yemeni companies, there is a different institutional and operational structure than for funds invested primarily in more developed markets. This indicates that investors in Yemen cannot expect to use litigation as easily as in neighboring countries.

The need for corporate governance stems from the separation between ownership and management (Aguilera and Jackson 2003; Shleifer and Vishny 1997). This is achieved by joint-stock companies, as many shareholders seek to invest their money in these companies

with the aim of earning a profit. Some of them do not have sufficient experience to manage a company. Therefore, experienced managers are hired to manage these companies to achieve their profitability goals and conduct their daily business, and these managers are not the company owners.

Agency theory had a significant impact on the emergence of corporate governance (Hassan et al. 2019) and was the basis of the corporate governance idea. The theory developed in response to shifts in the forms of ownership, which led to important developments in the areas of control and performance measurement (Sanchez-Marin et al. 2011; Billett et al. 2017; Bens and Monahan 2004; Bonsall and Miller 2017; Han et al. 2014; Jensen and Meckling 1976; Bathala and Rao 1995). Agency theory focuses on the issue of conflict between the principal and the agent and holds that this conflict can be resolved through corporate governance mechanisms, since the agent does not always strive to meet the principal's objectives. Conflict occurs when the agent and the principal do not communicate with each other consistently and accurately. Generally, agency theory describes the conflict between a principal and an agent and indicates that the conflict can be resolved through corporate governance mechanisms (Nguyen et al. 2015), since the agent does not always work to achieve the principal's objectives.

Corporate governance has been extensively studied (Loyola and Portilla 2014; Core et al. 2008; Soepriyanto et al. 2021; Fosberg 2004; Setiawan et al. 2016; Rutherford and Buchholtz 2007; Chen and Liu 2013; Feng et al. 2020; Tian and Lau 2001; Ghabri 2022; Core et al. 1999). Those studies examined the principles proposed, concerns regarding shareholder rights, equal treatment of shareholders, disclosure and transparency, the duties and responsibilities of the board of directors, and stakeholder rights. A good corporate governance system is one of the most important factors in creating a good, correct, and effective investment environment. The purpose of corporate governance is to provide investors with the information they need and a strong communication system that works effectively and efficiently (OECD 2004).

**H₁.** *Corporate governance factors (ownership structure, disclosure, transparency, responsibility, accountability, performance assessment, risk management, and control) are positively related to investment decisions.*

### 2.4. Specific Factors Related to the Need to Reform Corporate Governance

In any country, the aim of the economic system is to fulfil the needs of society by producing goods and services in sufficient quantity and quality to meet people's needs; for this to occur, the groundwork must be prepared for establishing appropriate entities within the framework to fulfil these functions and responsibilities (Ghisellini et al. 2016; Littig and Grießler 2005; Debreu 1951). As a representation of the private sector, it can be seen as private projects or companies that aim to achieve profits by investing capital in opportunities to meet society's needs. Generally, a business entity is defined as an instrument for accumulating capital to produce, distribute, and invest in goods and services to generate profit for its shareholders. There are several types of businesses, including sole proprietorships, general partnerships, limited partnerships, joint ventures, joint stock corporations, limited liability companies, and companies limited by shares. Each company has different methods of management and ownership.

Corporate governance is a relatively recent economic term that refers to joint-stock companies being subjected to laws and regulations that impose monitoring and follow-up and ensure that these companies' data as well as their administrative and financial practices are characterized by the highest levels of disclosure and transparency to protect the rights of shareholders. Corporate governance, with its connotations of disclosure, transparency, and guaranteed shareholder rights, is considered as a mechanism (Hong et al. 2021). It raises the efficiency and performance of financial markets and maximizes the value of companies, which contributes to these markets becoming an engine for economic growth rather than a damper, and works to reduce the failure and stumbling of these companies (Guluma 2021; Klapper and Love 2004; Al-Matari et al. 2016).

**H₂.** *The characteristics of governed business entities (sole proprietorships, general partnerships, limited partnerships, joint ventures, joint-stock corporations, limited liability companies, and companies limited by shares) are positively related to investment decisions.*

Most of the financial markets started to consider obliging the listed companies on the stock exchange to commit to applying the rules of governance as a new condition of registration. By studying the capital markets, especially the developed ones, we find that all of them apply a corporate governance system to their listed companies, which is a prerequisite for the companies to be able to list their shares on these stock exchanges.

However, in emerging markets, we find that many of those markets do not apply such a system for several reasons, including that many necessary mechanisms are missing in those markets, which are in dire need of implementing them. These may be among the most important reasons for the weak efficiency of capital markets in emerging countries, which has prevented these markets from reaching any form of recognized efficiency, including weak efficiency.

A stock market is necessary for Yemen to support small- and medium-sized businesses and transform them into joint-stock corporations that can support the national economy and attract local and foreign capital (Al-Homaidi et al. 2020a; Al-Matari et al. 2016; Alobaidi et al. 2017; Alasbahi 2021). Yemen is a market full of risks with limited infrastructure in various sectors, and this is an important first step, but it could also turn out to be a turning point for large companies, which distribute risk by investing in many companies rather than just one, creating a high cost of capital. Professional associations and the media play a major role in raising awareness about corporate governance, which coincides with stock markets and the presence of joint-stock companies to protect the rights of shareholders.

Legally, corporate governance aims to maintain the contractual obligations between companies and other parties while reducing the negative implications. Furthermore, corporate governance ensures that the rights of all parties in a company are protected, including shareholders, boards of directors, and executive management, as well as other stakeholders (Connelly et al. 2010; Alawaqleh and Ali Almasria 2021). It is essential to have laws, legislation, bylaws, and rules regulating work in companies, which constitute the backbone of corporate governance and support the people who work for companies. The most important of these laws are corporate, capital markets, banking, labor, commercial, international accounting and auditing standards, tax law, consumer protection, environmental protection, and other relevant laws. These laws, regulations, and systems are considered safety valves to ensure that corporate governance principles are followed well and soundly (Gong et al. 2018). For such legislation to be valuable and effective, the control and supervisory aspects of supervisory authorities must be activated, financial statements should be prepared under international accounting and auditing standards, and attention should be paid to the disclosure and transparency of the level of risk to shareholders and stakeholders.

**H₃.** *Specific factors related to corporate governance (awareness of setting up a stock exchange, investor protection, awareness via media, educational awareness, and awareness through professional institutions) are positively related to investment decisions.*

*2.5. Corporate Governance and Investment in an Uncertain Environment*

Many scholars (Mihail and Dumitrescu 2021; Ghabri 2022; Utama et al. 2017) have determined what investors take into consideration when making investment decisions, especially in light of an economic system that is characterized by globalization and intense competition between companies and various institutions to enter local or international capital markets for investment and the extent of companies' commitment to applying the principles of governance, which is one of the basic criteria affecting their decisions. Institutions that apply these principles increase their competitiveness in the long run due to the fact of transparency in their transactions, accounting, and financial audit procedures and all of their operations, which will inevitably boost the confidence of local and foreign

investors alike, who will then invest in these companies, thus lowering the cost of capital and allowing for more stability of funding sources.

Disclosure and transparency standards in the context of proper application of the principles of governance would assist in preventing the occurrence of banking crises (Khanchel 2007; Mihail and Dumitrescu 2021; Farah et al. 2021; Mamatzakis and Bermpei 2015; Al Maeeni et al. 2022; Benmelech and Bergman 2018). Consequently, if companies are faced with a specific crisis, their commitment to maintaining high standards for disclosure of information related to their debts and obligations will allow them to follow the legal procedures in the event of bankruptcy or expropriation, providing fairness for creditors and other stakeholders should either of those occur (Kirkpatrick 2009). The lack of correct CG rules creates the opportunity for corruption to occur in institutions. The owners, who can be on the board of directors, managers, or executives, are considered members of the institutions themselves (Johnson et al. 2000). Corruption can be seen in the looting of the institution or public money at the expense of shareholders, creditors, and other stakeholders. In the global economy, or even at the international level, when governance practices are weakened, companies become more vulnerable to dire consequences that far exceed mere scandals and financial crises. Therefore, it has become clear that the practice of corporate governance determines, to a large extent, the fate of companies and of all economies in the era of globalization.

Many international studies (Hassan et al. 2019; Larcker et al. 2007; Kimbro and Xu 2016; Menshawy et al. 2021; Ramírez et al. 2022; Almasria 2018, 2022a) have indicated that there is a great relationship, especially at the level of emerging markets, between stock performance in terms of price trends and levels of return and the extent of companies' commitment to applying the standards and principles related to the concept of governance to ensure the success of corporate management in preserving and developing shareholders' rights.

A lack of corporate governance means an increase in the power of corruption, as there will be no one to resist it (Johnson et al. 2000; Chang et al. 1998; Morck et al. 2005). Due to the prevalence of irresponsibility and lack of commitment creating an environment of insecurity and uncertainty, increased ambiguity, and an inability to distinguish among the options offered for investment, there will be an increased sense of nihilism and inability to think, as workers become machines and their motivation to work disappears, and an increase in workers who disregard administrative directives and deviate from regulations.

**H$_4$.** *There is a reluctance among investors to invest in Yemen.*

### 3. Materials and Methods

*3.1. Sample*

Many recent studies have highlighted the importance of corporate governance from a variety of perspectives (Chen and Lin 2022; Askarzadeh et al. 2022; Ramírez et al. 2022; Ghabri 2022; Si Tayeb et al. 2022; Chung et al. 2022; Muhammad et al. 2022; Bimo et al. 2022; Farah et al. 2021; Al-Gamrh et al. 2020; Al-ahdal et al. 2020; dos Santos et al. 2019; Adel et al. 2019; Shahid and Abbas 2019). We conducted a survey, reaching out to a wide range of individuals at firms and corporations in Yemen who are certified public accountants (YECPAs). Their involvement made it possible to obtain a clear understanding, based on a standardized set of questions for a more accurate assessment, of the role of CG in investment decisions by current and prospective investors and, specifically, the importance of CG in Yemen. Their participation in the questionnaire and the use of a set of relevant questions allowed for a close realistic assessment of the role of CG in investment decisions compared to other approaches, where this was closely related to practice. This approach has also helped in developing a deeper understanding of the aspects of CG factors that matter most in Yemen and those that need reform, while investigating the causes of different investment decisions in an environment of uncertainty surrounding the current situation in Yemen. The study was designed to enable a targeted focus on Yemeni companies and investors based on the 460 licenses for YECPAs issued by the Ministry of Industry and Trade in 2021, which were selected as the sample. The final sample included 312 YECPAs

who responded and took part in the survey (Table 2). A questionnaire was distributed via Google Forms, as well as manually, due to the occurrence problems with the Internet in Yemen, because many companies do not use electronic survey forms. The survey was conducted in October and November 2021.

**Table 2.** Sample description.

| Description | Responses |
|---|---|
| Total received responses | 326 |
| Mismatched | −12 |
| Missed experience | −2 |
| **Final sample** | **312** |

Source: authors.

### 3.2. Measures

The aim of this study was to measure the variables mentioned by the 312 respondents and categorize them accordingly. Investment decisions in governed companies were administered as a dependent variable scale (4 items) to assess how much people are willing to invest in the companies. After an exhaustive literature review, the objectives of the study in terms of independent variables related to CG were achieved: aspects of CG factors (7 items), awareness factors (6 items), and types of governed companies (7 items), on the five-point Likert scale, ranging from "strongly disagree" to "strongly agree". The survey also included factors related to investors' priorities when making the decision to invest money in an uncertain environment (5 items). The first draft of the survey was shared with the co-authors for evaluation against the study's objectives. The co-authors completed a trial version of the survey and provided feedback regarding its readability and usability in Supplementary Materials. The final version of the survey was modified both linguistically and technically until an agreement was reached, although the survey content did not change. Both Arabic and English versions were checked, although only the Arabic version was used for data collection. In addition, reliability analyses were conducted, and Cronbach's alpha ranged from 0.903 to 0.917 on the 25-item scale. The overall reliability of corporate governance factors, specific factors, and investment items was moderately high (26 items: $\alpha = 0.911$).

### 3.3. Data Analysis

Demographic data were collected based on a Likert scale. The responses of the YECPAs are expressed as percentages, and the statistical analysis was performed using Microsoft Excel spreadsheets and Stata. Pearson's linear correlation was used to assess the relationship between continuous variables, and frequency distribution was used to evaluate the relationship between categorical variables. We conducted multiple linear regression analyses in order to identify the factors that predict investment (INVEST). For all purposes, a *p*-value of 0.05 was considered statistically significant.

### 3.4. Econometrical Model

Equation (1) was used to investigate the relationship between corporate governance factors (CGFs), comprising ownership structure (OS), disclosure (DISC), transparency (TRAN), responsibility (RESP), accountability (ACC), performance assessment (PA), risk management (RM), control (CONT), and investment decisions (INVEST) using generalized least squares (GLS) regression:

$$\text{INVEST} = f\left(\sum \text{CGF}\right) + \varepsilon \qquad (1)$$

In the second step, we tested the impact of governed company entities (CGEs), comprising sole proprietorship (SOLE), general partnership (GP), limited partnership (LP), joint

venture (JV), joint-stock corporation (JSC), limited liability company (LLC), and company limited by shares (CLS) on INVEST using Equation (2):

$$\text{INVEST} = f\left(\sum \text{CGE}\right) + \varepsilon \tag{2}$$

Finally, we tested the impact of the specific factors of CG (CGSF), comprising awareness to set-up a stock market exchange (STK), awareness of transforming family business establishments and individual institutions into joint-stock corporations (TR), protecting investors pre- and post-investment (PROT), educational awareness of adopting CG (EDU), media awareness of adopting CG (MED), and awareness through relevant professional associations to adopt CG (PROF), on INVEST using Equation (3):

$$\text{INVEST} = f\left(\sum \text{CGSF}\right) + \varepsilon \tag{3}$$

## 4. Results

### 4.1. Demographic Characteristics of Participants

A total of 312 YECPAs participated in the study; 51.3% had a bachelor's degree, 30.8% had a master's degree, and 15.4% had a PhD. The majority of the participants majored in accounting (71.8%), followed by business administration (25.6%). In terms of experience, the sample was relatively evenly distributed between less than 5 years, 5 years to less than 10 years, and 10 years or more (35.9, 23.4, and 40.7%, respectively) (Table 3).

**Table 3.** Main characteristics of participants.

| Socio-Academic Characteristics | Total (%) |
|---|---:|
| **Academic degree** | |
| Bachelor's degree | 160 (51.3) |
| Postgraduate diploma | 8 (2.6) |
| Master's degree | 96 (30.8) |
| PhD | 48 (15. 4) |
| **Specialization** | |
| Accounting | 224 (71.8) |
| Business administration | 80 (25.6) |
| Finance | 8 (2.6) |
| **Years of experience** | |
| Less than 5 years | 112 (35.9) |
| 5 years to less than 10 years | 73 (23.4) |
| 10 years or more | 127 (40.7) |

### 4.2. Descriptive and Correlation Analysis

As shown in Table 4, joint-stock corporations had the highest mean for governed company entities at 3.56, followed by limited liability companies at 3.15. The highest means of CGFs were for DISC, TRANS, RESP, and ACC at 3.72 each, followed by OS and PA at 3.69 each. Overall, MED had the highest mean of specific CG factors at 3.74, indicating that it is a very important factor to mention when explaining how crucial CG is for investors.

As shown in Table 5, according to the Pearson correlation coefficient, there was a significant negative correlation ($-0.15$, *p*-value $< 0.05$) between INVEST and SOLE. Additionally, there was a positive correlation between INVEST (0.183, *p*-value $< 0.01$) and GP and no correlation between LP or JV and INVEST. However, the rest of the variables were positively correlated with INVEST (*p*-value $< 0.001$).

**Table 4.** Descriptive statistics.

| | | | | Shapiro–Wilk (Normality) | |
|---|---|---|---|---|---|
| **Variable** | **N** | **Mean** | **SD** | **W** | ***p*** |
| INVEST | 312 | 3.56 | 0.361 | 0.895 | <0.001 |
| SOLE | 312 | 2.31 | 0.462 | 0.58 | <0.001 |
| LP | 312 | 2.64 | 0.531 | 0.694 | <0.001 |
| JSC | 312 | 3.56 | 0.546 | 0.677 | <0.001 |
| CLS | 312 | 2.34 | 0.717 | 0.809 | <0.001 |
| GP | 312 | 2.34 | 0.68 | 0.802 | <0.001 |
| JV | 312 | 2.62 | 0.666 | 0.755 | <0.001 |
| LLC | 312 | 3.15 | 0.58 | 0.748 | <0.001 |
| OS | 312 | 3.69 | 0.515 | 0.598 | <0.001 |
| DISC | 312 | 3.72 | 0.505 | 0.574 | <0.001 |
| TRANS | 312 | 3.72 | 0.505 | 0.574 | <0.001 |
| RESP | 312 | 3.72 | 0.451 | 0.563 | <0.001 |
| ACC | 312 | 3.72 | 0.505 | 0.574 | <0.001 |
| PA | 312 | 3.69 | 0.515 | 0.598 | <0.001 |
| RM | 312 | 3.51 | 0.595 | 0.707 | <0.001 |
| CONT | 312 | 3.67 | 0.524 | 0.619 | <0.001 |
| STK | 312 | 3.69 | 0.515 | 0.598 | <0.001 |
| TR | 312 | 3.51 | 0.501 | 0.636 | <0.001 |
| PROT | 312 | 3.72 | 0.451 | 0.563 | <0.001 |
| EDU | 312 | 3.72 | 0.451 | 0.563 | <0.001 |
| MED | 312 | 3.74 | 0.437 | 0.544 | <0.001 |
| PROF | 312 | 3.64 | 0.531 | 0.637 | <0.001 |

Note. INVEST, investment decisions in governed companies; SOLE, sole proprietorships adopting corporate governance; LP, limited partnerships adopting corporate governance; JSC, joint-stock corporations adopting corporate governance; CLS, companies limited by shares adopting corporate governance; GP, general partnerships adopting corporate governance; JV, joint ventures adopting corporate governance; LLC, limited liability companies adopting corporate governance; OS, ownership structure; DISC, disclosure; TRANS, transparency; RESP, responsibility; ACC, accountability; PA, performance assessment; RM, risk management; CONT, control; STK, awareness of setting up stock market exchange; TR, awareness of transforming family businesses, establishments, and individual institutions into joint-stock corporations; PROT, protecting investors' pre- and post-investment; EDU, educational awareness of adapting CG; MED, media awareness of adapting CG; PROF, awareness through relevant professional associations of adopting CG.

*4.3. Hypothesis Test*

Table 6 presents the regression results from Equation (1). Among all corporate governance factors (CGFs), the estimated coefficients were positive and significant at $p < 0.001$. According to our expectations, higher CGF values are associated with increased investment decisions. Our findings support hypothesis $H_1$.

Once investors make their investment decisions, they may also look at the company's relationship with its stakeholders. We found out that companies that engage with and respond to a wide range of stakeholders are more sustainable and better prepared to achieve higher financial results. According to Coombes and Watson (2000), 80% of institutional and private equity owners are willing to pay a premium for these companies. Furthermore, the company's risks will be reduced by the integrity of the CG. As a result, it is plausible to suppose that well-governed companies have lower capital expenses than those that are not thought to be well-governed (Vernikov 2013). Therefore, stakeholders are a pertinent concern for all companies, whether the company's motivation is to serve the community or raise shareholder value, or both. With regard to how this applies to CG, the achievement of profitability and sustainability is contingent upon the board's ability to balance stakeholder interests with company objectives.

**Table 5.** Correlation matrix.

| | INVEST | SOLE | GP | LP | JV | JSC | LLC | CLS | OS | DISC | TRANS | RESP | ACC | PA | RM | CONT | TR | STK | PROT | EDU | MED | PROF |
|---|---|---|---|---|---|---|---|---|---|---|---|---|---|---|---|---|---|---|---|---|---|---|
| INVEST | — | | | | | | | | | | | | | | | | | | | | | |
| SOLE | −0.15 * | — | | | | | | | | | | | | | | | | | | | | |
| GP | 0.183 ** | 0.572 *** | — | | | | | | | | | | | | | | | | | | | |
| LP | 0.008 | 0.346 *** | 0.29 *** | — | | | | | | | | | | | | | | | | | | |
| JV | 0.066 | 0.386 *** | 0.34 *** | 0.627 *** | — | | | | | | | | | | | | | | | | | |
| JSC | 0.357 *** | 0.024 | 0.42 *** | 0.257 *** | 0.103 | — | | | | | | | | | | | | | | | | |
| LLC | 0.265 *** | 0.303 *** | 0.39 *** | 0.263 *** | 0.487 *** | 0.46 *** | — | | | | | | | | | | | | | | | |
| CLS | 0.199 *** | 0.31 *** | 0.42 *** | 0.54 *** | 0.7 *** | 0.33 *** | 0.43 *** | — | | | | | | | | | | | | | | |
| OS | 0.338 *** | 0.183 *** | 0.37 *** | 0.253 *** | 0.104 | 0.62 *** | 0.33 *** | 0.28 *** | — | | | | | | | | | | | | | |
| DISC | 0.584 *** | 0.042 | 0.15 ** | 0.101 | 0.059 | 0.58 *** | 0.41 *** | 0.21 *** | 0.655 *** | — | | | | | | | | | | | | |
| TRANS | 0.372 *** | 0.153 ** | 0.3 *** | 0.101 | 0.059 | 0.58 *** | 0.41 *** | 0.36 *** | 0.754 *** | 0.8 *** | — | | | | | | | | | | | |
| RESP | 0.456 *** | 0.171 ** | 0.17 ** | 0.006 | 0.066 | 0.54 *** | 0.36 *** | 0.16 ** | 0.512 *** | 0.78 *** | 0.67 *** | — | | | | | | | | | | |
| ACC | 0.514 *** | 0.153 ** | 0.3 *** | 0.101 | 0.059 | 0.49 *** | 0.24 *** | 0.21 *** | 0.655 *** | 0.7 *** | 0.7 *** | 0.78 *** | — | | | | | | | | | |
| PA | 0.546 *** | 0.291 *** | 0.44 *** | 0.159 ** | 0.254 *** | 0.35 *** | 0.33 *** | 0.28 *** | 0.515 *** | 0.66 *** | 0.66 *** | 0.73 *** | 0.85 *** | — | | | | | | | | |
| RM | 0.251 *** | 0.079 | 0.25 *** | −0.07 | −0.22 *** | 0.45 *** | 0.29 *** | 0.12 * | 0.601 *** | 0.66 *** | 0.74 *** | 0.64 *** | 0.57 *** | 0.52 *** | — | | | | | | | |
| CONT | 0.238 *** | 0.319 *** | 0.36 *** | 0.031 | 0.074 | 0.39 *** | 0.42 *** | 0.27 *** | 0.572 *** | 0.71 *** | 0.81 *** | 0.69 *** | 0.71 *** | 0.67 *** | 0.72 *** | — | | | | | | |
| TR | 0.334 *** | 0.205 *** | 0.3 *** | −0.08 | −0.02 | 0.35 *** | 0.35 *** | 0.14 * | 0.215 *** | 0.37 *** | 0.37 *** | 0.53 *** | 0.37 *** | 0.41 *** | 0.5 *** | 0.56 *** | — | | | | | |
| STK | 0.615 *** | −0.03 | 0.07 | −0.12 * | −0.05 | 0.35 *** | 0.25 *** | 0.07 | 0.321 *** | 0.66 *** | 0.46 *** | 0.62 *** | 0.36 *** | 0.42 *** | 0.6 *** | 0.38 *** | 0.51 *** | — | | | | |
| PROT | 0.456 *** | −0.08 | 0.08 | 0.006 | −0.11 | 0.44 *** | 0.07 | 0.08 | 0.512 *** | 0.55 *** | 0.55 *** | 0.49 *** | 0.44 *** | 0.4 *** | 0.45 *** | 0.36 *** | 0.42 *** | 0.62 *** | — | | | |
| EDU | 0.536 *** | −0.08 | 0 | −0.21 *** | −0.19 *** | 0.23 *** | 0.07 | −0.08 | −0.04 | 0.22 *** | 0.1 | 0.49 *** | 0.33 *** | 0.29 *** | 0.06 | 0.15 * | 0.53 *** | 0.4 *** | 0.37 *** | — | | |
| MED | 0.502 *** | 0.01 | 0.17 ** | 0.045 | −0.16 ** | 0.39 *** | 0.06 | 0.16 ** | 0.334 *** | 0.37 *** | 0.49 *** | 0.42 *** | 0.49 *** | 0.45 *** | 0.41 *** | 0.3 *** | 0.25 *** | 0.33 *** | 0.54 *** | 0.55 *** | — | |
| PROF | 0.377 *** | 0.032 | 0.21 *** | 0.089 | −0.17 ** | 0.43 *** | 0.01 | 0.14 * | 0.253 *** | 0.2 *** | 0.29 *** | 0.22 *** | 0.29 *** | 0.25 *** | 0.18 ** | 0.12 * | 0.4 *** | 0.16 ** | 0.54 *** | 0.54 *** | 0.8 *** | — |

INVEST, investment decisions in governed companies; SOLE, sole proprietorships adopting corporate governance; LP, limited partnerships adopting corporate governance; JSC, joint-stock corporations adopting corporate governance; CLS, companies limited by shares adopting corporate governance; GP, general partnerships adopting corporate governance; JV, joint ventures adopting corporate governance; LLC, limited liability companies adopting corporate governance; OS, ownership structure; DISC, disclosure; TRANS, transparency; RESP, responsibility; ACC, accountability; PA, performance assessment; RM, risk management; CONT, control; STK, awareness of setting up stock market exchange; TR, awareness of transforming family businesses, establishments, and individual institutions into joint-stock corporations; PROT, protecting investors pre- and post-investment; EDU, educational awareness of adopting CG; MED, media awareness of adapting CG; PROF, awareness through relevant professional associations of adopting CG. Standard errors in parentheses; * $p < 0.05$, ** $p < 0.01$, and *** $p < 0.001$.

**Table 6.** CG factors and investor decisions.

| Model | (1) | (2) | (3) | (4) | (5) | (6) | (7) | (8) |
|---|---|---|---|---|---|---|---|---|
| Variable | Invest | Invest | Invest | Invest | Invest | Invest | Invest | Invest |
| OS | 0.237 *** | | | | | | | |
| | (0.103) | | | | | | | |
| DISC | | 0.418 *** | | | | | | |
| | | (0.033) | | | | | | |
| TRANS | | | 0.266 *** | | | | | |
| | | | (0.038) | | | | | |
| RESP | | | | 0.365 *** | | | | |
| | | | | (0.040) | | | | |
| ACC | | | | | 0.367 *** | | | |
| | | | | | (0.035) | | | |
| PA | | | | | | 0.382 *** | | |
| | | | | | | (0.033) | | |
| RM | | | | | | | 0.153 *** | |
| | | | | | | | (0.033) | |
| CONT | | | | | | | | 0.164 *** |
| | | | | | | | | (0.038) |
| Constant | 1.920 *** | 1.422 *** | 1.834 *** | 1.565 *** | 1.560 *** | 1.528 *** | 2.174 *** | 2.120 *** |
| | (0.103) | (0.901) | (0.104) | (0.111) | (0.096) | (0.091) | (0.086) | (0.103) |
| Observations | 312 | 312 | 312 | 312 | 312 | 312 | 312 | 312 |
| R | 0.338 | 0.584 | 0.372 | 0.456 | 0.514 | 0.546 | 0.251 | 0.238 |
| $R^2$ | 0.114 | 0.342 | 0.139 | 0.208 | 0.264 | 0.298 | 0.063 | 0.057 |

INVEST, investment decisions in governed companies; OS, ownership structure; DISC, disclosure; TRANS, transparency; RESP, responsibility; ACC, accountability; PA, performance assessment; RM, risk management; CONT, control; STK, awareness of setting up stock market exchange; TR, awareness of transforming family businesses, establishments, and individual institutions into joint-stock corporations; PROT, protecting investors pre- and post-investment; EDU, educational awareness of adopting CG; MED, media awareness of adopting CG; PROF, awareness through relevant professional associations of adapting CG. Standard errors in parentheses; *** $p < 0.001$.

Table 7 shows the regression results from Equation (2). The coefficient of sole proprietorships (SOLE) was significantly negative at $p < 0.05$. However, the findings indicate significantly positive coefficients for general partnerships (GP) at $p < 0.01$, joint-stock corporations (JSC) at $p < 0.001$, limited liability companies at $p < 0.001$, and companies limited by shares (CLS) at $p < 0.001$. Therefore, investors in Yemen prefer to invest in JSC, because most companies in Yemen that adopt CG are joint-stock companies. The coefficients for limited partnerships (LP) and joint ventures (JV) were positive and statistically insignificant, indicating that those entities are not associated with investors' options during the investment decision-making process. The findings partially confirm hypothesis H$_2$.

Table 8 lists the stock exchanges of West Asian countries, except for Yemen, as it does not yet have a stock market. The stock market provides good motivation for many companies and obliges them to apply the principles of CG, thus increasing investor confidence. This contributes to encouraging and attracting segments of society to shift to a corporate environment and subsequently encourages the establishment of a stock market similar to those of neighboring countries and other countries in the world. As a result, this will attract current and prospective investors while highlighting the importance of CG to the pursuit of sustainable development for companies by providing guidelines, regulations, and systems for the members of boards of directors, avoiding risks related to the performance and management of companies and consolidating the concept of good CG. This will lead to companies achieving their goals efficiently and effectively with the optimum use of resources and conscious control of risks and developing an intelligent strategy for their business.

**Table 7.** Governed corporate entities and investment decisions.

| Model | (1) | (2) | (3) | (4) | (5) | (6) | (7) |
|---|---|---|---|---|---|---|---|
| Variable | Invest | Invest | Invest | Invest | Invest | Invest | Invest |
| SOLE | −0.113 * | | | | | | |
| | (0.044) | | | | | | |
| GP | | 0.097 ** | | | | | |
| | | (0.030) | | | | | |
| LP | | | 0.005 | | | | |
| | | | (0.039) | | | | |
| JV | | | | 0.036 | | | |
| | | | | 0.031 | | | |
| JSC | | | | | 0.236 *** | | |
| | | | | | (0.035) | | |
| LLC | | | | | | 0.165 *** | |
| | | | | | | (0.034) | |
| CLS | | | | | | | 0.100 *** |
| | | | | | | | (0.028) |
| Constant | 2.706 *** | 2.363 *** | 2.550 *** | 2.500 *** | 1.954 *** | 2.203 *** | 2.358 *** |
| | (0.061) | (0.063) | (0.067) | (0.054) | (0.0919) | (0.076) | (0.060) |
| Observations | 312 | 312 | 312 | 312 | 312 | 312 | 312 |
| R | 0.145 | 0.183 | 0.008 | 0.067 | 0.357 | 0.265 | 0.199 |
| $R^2$ | 0.021 | 0.034 | 0.000 | 0.004 | 0.127 | 0.070 | 0.040 |

INVEST, investment decisions in governed companies; SOLE, sole proprietorships adopting corporate governance; LP, limited partnerships adopting corporate governance; JSC, joint-stock corporations adopting corporate governance; CLS, companies limited by shares adopting corporate governance; GP, general partnerships adopting corporate governance; JV, joint ventures adopting corporate governance; LLC, limited liability companies adopting corporate governance. Standard errors in parentheses; * $p < 0.05$, ** $p < 0.01$, *** $p < 0.001$.

**Table 8.** Stock exchanges in West Asia.

| Economy | Exchange | Location | Founded |
|---|---|---|---|
| Bahrain | Bahrain Stock Exchange | Manama | 1987 |
| Iran | Tehran Stock Exchange | Tehran | 1967 |
| | Iran Fara Bourse | Tehran | 2008 |
| | Iran Mercantile Exchange | Tehran | 2006 |
| | Iranian Energy Exchange | Tehran | 2008 |
| Iraq | Iraq Stock Exchange | Baghdad | 2004 |
| Israel | Tel Aviv Stock Exchange | Tel Aviv | 1953 |
| Jordan | Amman Stock Exchange | Amman | 1999 |
| Kuwait | Boursa Kuwait | Safat | 1977 |
| Lebanon | Beirut Stock Exchange | Beirut | 1920 |
| Oman | Muscat Securities Market | Muscat | 1988 |
| Palestine | Palestine Securities Exchange | Nablus | 1995 |
| Qatar | Doha Securities Market | Doha | 1997 |
| Saudi Arabia | Tadawul | Riyadh | 2007 |
| Syria | Damascus Securities Exchange | Damascus | 2009 |
| United Arab Emirates | Abu Dhabi Securities Market | Abu Dhabi | 2000 |
| | Dubai Financial Market | Dubai | 2000 |
| | NASDAQ Dubai | Dubai | 2005 |
| | Dubai Gold & Commodities Exchange | Dubai | 2005 |

Table 9 shows the coefficients for the awareness of transforming family businesses and individual institutions into joint-stock corporations (TR). Protecting investors pre- and post-investment (PROT), educational awareness of adopting CG (EDU), media awareness of adopting CG (MED), and awareness through relevant professional associations of adopting CG (PROF) were significantly positive at $p < 0.001$. Overall, the survey participants stressed the importance of raising awareness about converting family companies, institutions, and individual institutions into joint-stock companies and indicated 100% approval for converting family and individual companies into joint-stock companies. This emphasizes the importance of striving to carry out such transformation, because joint-stock corporations are better governed. As a result, these findings confirm hypothesis $H_3$.

**Table 9.** Specific factors of CG and investor decisions.

| Model | (1) | (2) | (3) | (4) | (5) | (6) |
|---|---|---|---|---|---|---|
| Variable | Invest | Invest | Invest | Invest | Invest | Invest |
| TR | 0.241 *** | | | | | |
| | (0.039) | | | | | |
| PROT | | 0.356 *** | | | | |
| | | (0.040) | | | | |
| STK | | | 0.431 *** | | | |
| | | | (0.031) | | | |
| EDU | | | | 0.429 *** | | |
| | | | | (0.038) | | |
| MED | | | | | 0.414 *** | |
| | | | | | (0.041) | |
| PROF | | | | | | 0.256 *** |
| | | | | | | (0.036) |
| Constant | 1.953 *** | 1.565 | 1.397 *** | 1.393 *** | 1.422 *** | 1.882 *** |
| | (0.099) | (0.111) | (0.086) | (0.106) | (0.113) | (0.096) |
| Observations | 312 | 312 | 312 | 312 | 312 | 312 |
| R | 0.334 | 0.456 | 0.615 | 0.536 | 0.502 | 0.377 |
| $R^2$ | 0.112 | 0.208 | 0.379 | 0.287 | 0.252 | 0.142 |

INVEST, investment decisions in governed companies; TR, awareness of transforming family businesses, establishments, and individual institutions into joint-stock corporations; PROT, protecting investors pre- and post-investment; EDU, educational awareness of adopting CG; MED, media awareness of adopting CG; PROF, awareness through relevant professional associations of adopting CG. Standard errors in parentheses; *** $p < 0.001$.

Moreover, investor protection pre- and post-investment by some companies would contribute to good governance and the expansion of investments in other companies. Shareholders and investors will be confident only if they are assured that they will receive fair and equal treatment, whether they are local citizens or foreigners. Therefore, an effective CG system must provide tools that shareholders can use to protect their rights and the ability to bring lawsuits against directors and members of the board of directors. However, it seems essential to explore this question, because CG, as a form of investor protection, may be expected to influence investor behavior; this will be critical to understanding the role of CG. This is especially the case in the context of CG in Yemen, where many institutions may not fully protect investor rights. Good governance can provide appropriate laws, rules, ownership rights, and means to resist corruption (Omri and Bel Hadj 2020).

The lack of awareness of the importance of CG in Yemen contributed to the situation of many investors avoiding investing their money or investing it less in new or existing companies, and this led to the establishment of a stock exchange being overlooked. Furthermore, there was no doubt among the survey respondents that educational awareness related to CG at the undergraduate and postgraduate levels in related disciplines should enhance acceptance of CG. We suggest that awareness campaigns be explicitly directed to several groups, including members of corporate boards, senior officials of public joint-stock corporations, CG and investor relations officers, potential shareholders and investors, and those with a general interest, such as researchers and media professionals. Many resources

should be used, including social networks, satellite channels, newspapers, websites, professional institutes and associations, and universities. Musleh Al-Sartawi and Sanad (2019) pointed out the necessity of offering workshops and training courses to raise awareness of the application of CG and compliance with CG law in Bahrain.

*4.4. Robustness of Investment Decisions in an Uncertain Environment*

In this section, we try to answer the fourth research question concerning why investors refrain from investing and establishing companies but invest in real estate. We also try to strengthen our previous findings. The private sector in Yemen faces many challenges in the business and investment environment. These challenges escalated with the continuation of the current conflict, to the extent that Yemen found itself at the bottom of the list of global business indicators, often occupying last place. As a result, businessmen in many parts of the country decided to move their capital to places outside Yemen. In contrast, those who decided to stay had to sacrifice large portions of their workforce. This is consistent with our survey. Figure 1 shows that 53.91% of investors directed their investments toward real estate, 22.98% invested abroad, 2.57% invested in personal savings and bank deposits, and the rest continued to invest in private projects. Our results support hypothesis $H_4$.

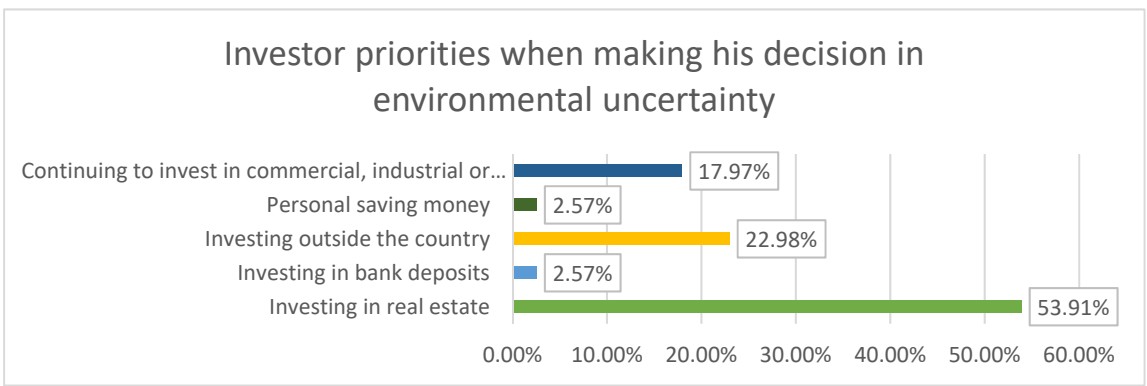

**Figure 1.** Investor priorities when making decisions amid environmental uncertainty.

By implementing CG, companies can increase shareholders' trust and confidence in their investments, because it indicates that the board of directors and executive management are aware of the risks the company faces and are able to manage and reduce these risks, which helps investors to make investment decisions in consideration of other essential investment criteria (Fooladi and Farhadi 2017). This is because CG is an effective practice that leads to attracting investors and gaining their confidence because of the advantages to the company, the most important of which is providing fairness and transparency for all stakeholders. There is also the potential for broader macroeconomic consequences, such as contagion and investment impacts on implementation. This was evident in the financial crisis that began in mid-2007 and the fact that poor CG can lead to markets losing confidence in companies' ability to properly manage their assets and liabilities, including during a liquidity crisis. Accordingly, good CG provides appropriate incentives for the board of directors and management to achieve goals that are in the best interest of the company and its shareholders, and it can facilitate effective monitoring within the company (Chen et al. 2012). This also supports the importance of CG in mitigating the impact of the agency problem on managers' cost adjustment decisions.

CG is essential to achieving and maintaining public confidence in the sustainability of companies and the economy and is critical to the proper operation of companies. Good governance has long been considered essential to enhance the long-term value of stakeholders in the business environment (Cohen et al. 2002), where good CG is more than just good practice, it is an indispensable component of any market. Accountability by corporate boards will increase due to the demands from investors and others, further enhancing the

quality of management oversight and ultimately leading to more efficient markets. Poor management of companies can contribute to failure, which can cause significant overhead costs, and failure can affect any viable system.

## 5. Discussion

Our survey highlighted ownership structure, disclosure, transparency, responsibility, accountability, performance evaluation, risk management, and control as factors of CG. The findings show the degree of importance of CGFs for investors. This is consistent with the results of (McKinsey & Company 2002), which surveyed institutional investors and shareholders with a focus on emerging markets and showed that 80% of investors were willing to pay a premium for well-governed companies (Klapper and Love 2004). In their study of 495 companies located in 25 emerging countries, the researchers found that good governance was associated with better performance and market valuation. They also concluded that CG is more important for companies located in countries with low shareholder protection and weak judicial systems. In our study, 100% of the respondents agreed that well-governed companies overcome the environmental uncertainty of investment, which is a significant percentage. Yu et al. (2017) concluded that an increase in the value of firms under good governance occurs only in competitive industries. On the other hand, we found that governed companies encourage investment, which leads to the sustainable development of companies.

Consistent with Shahid and Abbas (2019), we found that transparency and accountability are needed to promote long-term investment, financial stability, and business integrity, which supports stronger growth and a more inclusive society. Undoubtedly, enhancing transparency and disclosure is a prerequisite for exchanges, especially in terms of attracting investors and reassuring them of the market's integrity. Based on their function as an information resource for the investor community, stock exchanges often play a greater role in facilitating corporate disclosure than promoting other CG practices. As in other areas, the role of stock exchanges in monitoring corporate disclosure is often shared with securities regulators (Amico 2012). In addition, consistent with Leuz et al. (2009) and O'Connor and Byrne (2015), ownership structure, performance assessment, and control play significant roles in the success of CG, allowing investors a greater role in monitoring, which strengthens the company. The stock value, or the discount of companies with good CG, rises in many markets. For example, Farooque et al. (2010) indicated that ownership concentration and firm performance have a mutual influence on each other.

Many studies have been conducted on how decision makers can suitably adjust governance mechanisms to increase the performance of companies (Leuz and Wysocki 2016; Black et al. 2015; Ellili 2022; Almaqtari et al. 2021; Al-Gamrh et al. 2020; Ararat et al. 2021; Ararat et al. 2017). The implementation of robust CG mechanisms reduces investors' costs of controlling activity (Bushee and Noe 2000). In their survey, McCahery et al. (2016) concluded that CG is a vital aspect of institutional investment decisions for investors. The results also showed that the type of institutional investor affects the preference for governance mechanisms. Research conducted by McKinsey & Company showed that investors are willing to pay a higher price per share for companies with good CG practices (Coombes and Watson 2000). The higher share value is generally lower based on the investor's view of the market's maturity compared to companies without such practices. Improving CG by increasing transparency and access to information is essential to attract more investors and create greater opportunities to encourage companies to go public. Investing in companies with strong shareholder protection will increase, because investors often exclude expropriation.

The MENA region is characterized by family businesses, businesses owned by royal families, or state ownership (Farah et al. 2021). The largest sectors in Yemen are based on family businesses and state-owned companies. Large institutions have collapsed due to the fact of disagreements between owners and founders. Furthermore, in MENA countries, there are significant CG gaps in terms of institutional rules and enforcement, leading

to high rates of corruption and economic instability in the region (Aguilera et al. 2019). Implementing CG helps companies create a sound work environment that allows them to achieve better performance with good management; therefore, their economic value is more significant. In addition, good CG helps companies access financial markets and obtain the necessary financing at a lower cost, which allows them to expand their activity, reduce risks, and build trust with stakeholders. Although family businesses are significant to national economies, their future is fraught with risks, challenges, and difficulties that threaten their sustainability and transmission between generations. According to international reports, 95% of them will disappear and not extend beyond the third generation; according to the Institute of Family Businesses, only 30% of family businesses remain until the second generation, 12% until the third generation, and only 3% until the fourth generation and beyond.

The legal system in each country defines the responsibilities that corporations may have to shareholders and other relevant stakeholders, with awareness of the challenges and risks of globalization and the mechanisms and methods of work in light of the mechanisms of the market economy, by setting guidelines, rules, and laws to protect investors and ensure the growth of the economy (Shleifer and Vishny 1997; OECD 2004; Klapper and Love 2004; Farah et al. 2021; dos Santos et al. 2019; Aguilera et al. 2015). This is consistent with Ghabri (2022), who suggested that the institutional and legal environment is critical in determining the maximum level of good governance practices. Our findings are also consistent with those of Dahlquist et al. (2003), Bushee et al. (2014), and Leuz et al. (2009), who found that well-governed companies are important to institutional investors even in an environment with weak legal protection.

In response to our survey, 97.43% of respondents said companies could compete by adopting good governance if a stock market was established. The absence of CG in the business community is considered an economic imbalance. It is directly reflected in the productive structure of the national economy. This absence has led to a weak ability to attract private investment and low competitiveness in the private sector. One of the most important reasons why a CG system in Yemen has not been implemented may be the lack of a stock exchange market, which would include the conditions for registering institutions and companies; Yemen is the only country in the Middle East that does not have a stock exchange. CG can only occur by properly implementing its stages and developing a strategy for companies that implement it. Among the respondents, 2.57% disagreed, which can be explained by the fact that some participants looked at the situation that the country is going through with despair, believing that establishing a stock exchange in Yemen will be impossible.

CG provides a degree of confidence that the market economy will function properly. This is consistent with a study of Islamic banks by Nawaz (2019), who found that a robust governance mechanism increases market value. Moreover, it provides guidance for the board of directors, including how to set the company's strategies and objectives, determine its risk tolerance and the general management of its day-to-day business, protect the interests of depositors, and fulfil shareholders' obligations while taking into account their interests. Other recognized stakeholders align with the company's activities and behavior with the expectation that the company will operate safely and soundly with integrity and under applicable laws and regulations. The main challenge these organizations face is the lack of access to capital, financial resources, and investments. For most start-ups and MSMEs, identifying the right high-value investors and seeking advice on business growth are almost always the main priorities, while they are unaware that a strong commitment to CG is essential for the development process.

## 6. Policy Implications and Recommendations

The study findings have many implications in terms of policies and recommendations for disseminating awareness of CG and applying it, especially the adoption of disclosure and transparency by companies, which help to create confidence among investors. The

findings also shed light on the importance of seeking to establish a stock exchange market in Yemen. In addition, this study's policy lessons may apply to smaller countries with similar features and problems as Yemen. The findings also have important implications for those who manage and control companies, who need to analyze and think critically about these determinants in order to design strategies and mechanisms that will effectively support companies and investors and maintain their market viability. In the context of a lack of confidence in the legal system based on preserving the rights of investors, raising awareness of the application of governance can help guide investment in joint-stock companies and the transformation of nongoverned family companies in their various legal forms, and restore confidence. Finally, these findings have important implications for academics and researchers in the region, as they pave the way for further investigation.

Finally, the principles of corporate governance, if properly implemented, represent a way forward for individuals, institutions, and society as a whole, because it will provide individuals with an appropriate guarantee of achieving reasonable profits for their investments. Good corporate governance reduces risks, stimulates performance, improves access to capital markets, and improves the ability to market products and services, ensuring greater transparency and social accountability. Applying governance regulations would limit the occurrence of crises in the financial market, as experience and studies have shown that most of the crises that affect financial and banking institutions occur because they lack a framework for corporate governance; in addition, corporate governance would improve the performance and efficiency of financial markets by way of economic and financial guidance that works to reduce the occurrence of financial crises in these markets.

Based on our current findings, we recommend the following: It is necessary to enhance the presence of regulatory bodies to regulate companies in a more effective way than what is currently the case, which would help to strengthen the financial market. The principles of efficiency, responsibility, and transparency should be adopted, and companies should work based on market mechanisms to increase the performance and effectiveness of the financial market. The role of the media within the framework of governance should be enhanced, especially with regard to revitalizing the stock market. The application of international standards of governance should be ensured, and the pace of reforms that would raise the performance and efficiency of the financial market should be accelerated. A system of corporate governance based on rules and not relationships should be established. Companies should work to establish a strong corporate governance structure and adopt corporate governance charters. Furthermore, future research should be conducted to investigate how transparency, accountability, and independence affect economic development to determine which has the greatest impact on the relevant countries. Moreover, it is hoped that this study will make a significant contribution to the field that will pave the way for further research and studies.

## 7. Conclusions

The current study investigated the importance of CG and its impact on investors and investee companies in Yemen by surveying a sample of 312 certified public accountants. The survey, which was conducted during October and November 2021, included questions about the crucial factors of CG and their impact on investors' decisions, along with some other related factors. The results indicate that CG is important for companies, as it helps to attract investors and increase investor confidence and helps to increase disclosure and transparency in financial reports.

The findings support the need for greater awareness by regulators, policymakers, and standard-setters to protect the rights of company shareholders by providing seminars and courses for the media, unions, and professional associations to introduce the importance of CG practices in Yemen. Our findings indicate that all of the corporate governance factors (CGFs) had positive coefficients that were statistically significant at $p < 0.001$. This suggests that higher CGF values are associated with more favorable decisions regarding investments. Furthermore, the integrity of CG will enable companies to reduce their exposure to risks in

the future. A significant negative coefficient was found for sole proprietorships (SOLE) at a *p*-value of 0.05. However, the findings show significantly positive coefficients for general partnerships (GP) at $p < 0.01$, joint-stock corporations (JSC) at $p < 0.001$, limited liability companies (LLC) at $p < 0.001$, and companies limited by shares (CLS) at $p < 0.001$. Investors in Yemen prefer JSCs, since most companies that adopt CG are joint-stock companies.

This study's survey results clearly demonstrate that CG can lead to increased economic efficiency and improved financial security indicators. Further, the results also indicate that CG might be more important for developing countries with limited resources due to the fact of its association with the ability to attract the foreign investment necessary for development. On the other hand, there is a discrepancy in the factors of CG that are important to investors, and some reforms must be made to those related to the investment environment, such as establishing a stock market, improving investor protection laws, and raising awareness about the importance of CG. Therefore, it appears that the implementation of CG in Yemen should not be considered a luxury but rather a necessity imposed by the need to attract more investments, enhance investor confidence in local markets, and contribute to the country's financial soundness, especially since it appears from the investor opinion survey that CG has a more important role to play in developing countries with less mature financial markets such as Yemen.

This study contributes to the literature on CG and investors in Yemen. It provides useful insights and investigative evidence for auditors, managers, analysts, regulators, investors, academics, and other interested parties. The unique contribution of this study is the results of a survey on the importance of CG and its impact on investors and investee companies in Yemen. The study provides valuable theoretical and practical evidence. Accordingly, the research results can be useful for regulators as they assess whether CG achieves the desired goals and meets the need for improved quality of CG principles and factors. Moreover, the study highlights the level of compliance with CG principles in Yemen based on the importance of disclosure and transparency of financial data in companies and enhancing investor confidence. Thus, there is a need to establish a stock market in Yemen to supervise companies and stimulate their transformation into joint-stock companies.

**Supplementary Materials:** The following supporting information can be downloaded at: https://www.mdpi.com/article/10.3390/economies11010013/s1.

**Author Contributions:** Conceptualization, F.A., J.A., and N.A.A.; Data curation, F.A.; Formal analysis, F.A.; Funding acquisition, F.A. and N.A.A.; Investigation, F.A. and N.A.A.; Methodology, F.A. and N.A.A.; Project administration, F.A.; Software, F.A.; Supervision, F.A.; Validation, F.A., J.A. and N.A.A.; Visualization, F.A. and N.A.A.; Writing—original draft, F.A. and N.A.A.; Writing—review and editing, F.A., J.A. and N.A.A. All authors have read and agreed to the published version of the manuscript.

**Funding:** This research received no external funding.

**Informed Consent Statement:** Informed consent was obtained from all subjects involved in the study.

**Data Availability Statement:** The data presented in this study are available upon request from the first author. The data are not publicly available due to the presence of ethical restrictions.

**Acknowledgments:** The publication of this research was supported by the Deanship of Scientific Research and Graduate Studies at Philadelphia University, Jordan.

**Conflicts of Interest:** The authors declare no conflict of interest.

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
