# Peer review of "Examining the Impact of Corporate Governance on Investors and Investee Companies: Evidence from Yemen"

_economies, doi:10.3390/economies11010013_

Round 1

Reviewer 1 Report

I have carefully read the paper which Examining the impact of corporate governance on investors and investee companies: Evidence from Yemen and aimed to provide a robust solution through corporate governance. However, some revisions should be made before publication. The introduction section is written well and composed very well. However, a good intro presentation could be precise. The introduction needs to be more structured in terms of current citation from reputed journals.

Although the overall language of the manuscript is good and acceptable, few grammatic and typo issues have been observed.

Author(s) partially define the amount of literature that has been carried out and where does gap persist in this area of filed. Literature review is too short.

I suggest using summary on which you reviewed the literature. Also, it would be nice to summarize the main gaps in the literature that your paper should fill, because it should be clear. Better to add a summarizing paragraph at the end of the introduction. Overall, this Section contains enough explanation however the proposed model should be compared with some popular models.

The result and discussion section have been presented in good structure. Result section should be compared with current and relevant citations. Again, the closing paragraph of the result can add to conclude technical findings.

Authors are suggested to improve the conclusion section as well since it broadly handled and should be very concrete for the description of the results followed by the policy. I would suggest a separate policy section after conclusion.

I hope my comments would help to improve the paper before publication.

Author Response

Reply to Reviewer1

  1. The introduction section is written well and composed very well. However, a good intro presentation could be precise. The introduction needs to be more structured in terms of current citation from reputed journals.

Thank you for your comment. The paragraphs have been rearranged and some related paragraphs have been added.

  1. Although the overall language of the manuscript is good and acceptable, few grammatic and typo issues have been observed.

Thank you for your comments, We forwarded it to a language professional for proofreading.

  1. Author(s) partially define the amount of literature that has been carried out and where does gap persist in this area of filed. Literature review is too short. I suggest using summary on which you reviewed the literature. Also, it would be nice to summarize the main gaps in the literature that your paper should fill, because it should be clear. Better to add a summarizing paragraph at the end of the introduction. Overall, this Section contains enough explanation however the proposed model should be compared with some popular models.

Thank you for your comment. We added a summary in the introduction. In addition, we extended the literature review with many related studies.

  1. The result and discussion section have been presented in good structure. Result section should be compared with current and relevant citations. Again, the closing paragraph of the result can add to conclude technical findings.

Thank you for your comment. We added a discussion part to conclude the results of this study. Otherwise, we clarified the findings of the study.

  1. Authors are suggested to improve the conclusion section as well since it broadly handled and should be very concrete for the description of the results followed by the policy. I would suggest a separate policy section after conclusion.

Thank you for your comment. We added the policy section after the discussion section and improved the conclusion.

I hope my comments would help to improve the paper before publication.

Lastly, I would like to thank you for your time and effort in helping me improve this paper's quality. Thank you for your comments, and I hope my revisions and modifications will meet your expectations.

Reviewer 2 Report

Comments and Suggestions for Authors

Dear Author(s),

Thank for allowing me to read the manuscript.  The paper is well written and well structured. Please see below my detailed suggestions, to improve the paper
,
1. Title:
Examining the impact of corporate governance on investors and investee companies: Evidence from Yemen

The tile is good; it is precise and clearly state the purpose of the study.

2. Abstract
The abstract should be rewritten. The corporate governance (CG) is narrowly discussed in first sentence, it is not only the technique in enhancing investors’ confidence. Similarly, it lacks the proper flow, for example, “This study provides valuable insights for regulators, practitioners, and academicians on this issue. The survey is limited to a sample of chartered accountants. It would be useful to extend this survey to a larger sample, including a sample of supervisory managers in companies. It offers a unique contribution since it aims to clarify the importance of corporate governance for Yemeni investors and investee companies. Also, it is in quite detail, the highlighted components need to be removed or re-written”. The contribution of the paper is written in first sentence as well as in last sentence.  

3.Introduction
The introduction is good except some correction in terms of language and writing style.

4. Literature Review
It is good to see some changes as suggested, Now it fits for the purpose.

5.Methodology
This section also looks good after revision.

6.Results and Discussion
After the revision, the section also looks good.

7.Conclusion and Policy Implication
Now this section looks good too.

8.Overall Expression of the Report

This manuscript is good and can be accepted for publication after thoroughly proof reading it.

Author Response

Reply to Reviewer2

The tile is good; it is precise and clearly state the purpose of the study.

  1. Abstract
    The abstract should be rewritten. The corporate governance (CG) is narrowly discussed in first sentence, it is not only the technique in enhancing investors’ confidence. Similarly, it lacks the proper flow, for example, “This study provides valuable insights for regulators, practitioners, and academicians on this issue. The survey is limited to a sample of chartered accountants. It would be useful to extend this survey to a larger sample, including a sample of supervisory managers in companies. It offers a unique contribution since it aims to clarify the importance of corporate governance for Yemeni investors and investee companies. Also, it is in quite detail, the highlighted components need to be removed or re-written”. The contribution of the paper is written in first sentence as well as in last sentence.  

Thank you for your comment. We have made it more shareable.

3.Introduction
The introduction is good except some correction in terms of language and writing style.

Thank you for your comment. It was forwarded to a language professional for proofreading.

  1. Literature Review
    It is good to see some changes as suggested, Now it fits for the purpose.

5.Methodology
This section also looks good after revision.

6.Results and Discussion
After the revision, the section also looks good.

7.Conclusion and Policy Implication
Now this section looks good too.

8.Overall Expression of the Report
This manuscript is good and can be accepted for publication after thoroughly proof reading it.

Lastly, I would like to thank you for your time and effort in helping me improve this paper's quality. Thank you for your comments, and I hope my revisions and modifications will meet your expectations.

Reviewer 3 Report

Dear authors,

The paper is interesting; however, it suffers weakness as follows you need to revise the paper according to the comments:

1-    The abstract is too long and it should be reduced as much as possible especially numerical numbers.

2-    The theoretical issues and literature should be improved by the most recent studies, I listed the related studies as follow:

-Corporate governance and investment efficiency in Indonesia: the moderating role of industry competition

-The Effect of Corporate Governance Structure on Fraud and Money Laundering

-The impact of corporate governance measures on firm performance: the influences of managerial overconfidence

-Sustainability reporting and corporate reputation: the moderating effect of CEO opportunistic behavior

3- It is recommended to conduct robustness test on the study

Author Response

Reply to Reviewer3

The paper is interesting; however, it suffers weakness as follows you need to revise the paper according to the comments:

  • The abstract is too long and it should be reduced as much as possible especially numerical numbers.

Thank you for your comment. We have made it more shareable.

2-    The theoretical issues and literature should be improved by the most recent studies, I listed the related studies as follow:

-Corporate governance and investment efficiency in Indonesia: the moderating role of industry competition

-The Effect of Corporate Governance Structure on Fraud and Money Laundering

-The impact of corporate governance measures on firm performance: the influences of managerial overconfidence

-Sustainability reporting and corporate reputation: the moderating effect of CEO opportunistic behavior

Thank you for your suggestions. We have enriched the literature review with those you mentioned and others.

  • It is recommended to conduct robustness test on the study

Thank you for your comment. The last statistics findings are robust to what we argue in this paper. We added clarification of this.

Lastly, I would like to thank you for your time and effort in helping me improve this paper's quality. Thank you for your comments, and I hope my revisions and modifications will meet your expectations.

Round 2

Reviewer 3 Report

Dear authors,

You incorporated my comments on the paper, Thank you